# Effects of Diets Containing Finger Millet Straw and Corn Straw on Growth Performance, Plasma Metabolites, Immune Capacity, and Carcass Traits in Fattening Lambs

**DOI:** 10.3390/ani10081285

**Published:** 2020-07-28

**Authors:** Xiaoyong Chen, Hao Mi, Kai Cui, Rongyan Zhou, Shujun Tian, Leying Zhang

**Affiliations:** 1College of Animal Science and Technology, Hebei Agricultural University, Baoding 071001, China; rongyanzhou@126.com (R.Z.); tsj7890@126.com (S.T.); 2College of Life Science and Food Engineering, Hebei University of Engineering, Handan 056038, China; mihao137604660@163.com (H.M.); zhangly056000@126.com (L.Z.); 3Feed Research Institute, Chinese Academy of Agricultural Sciences, Beijing 100081, China; cuikai@caas.cn

**Keywords:** millet straw, corn straw, lamb, daily gain, blood metabolites, carcass traits

## Abstract

**Simple Summary:**

As the byproduct of crops, straw is an important roughage resource in sheep production in rural areas. Finger millet straw crop has widely been planted in arid and semi-arid area of the world. However, little is known about the effect of mixed diets containing millet straw and corn straw on the production of fattening lambs. This research evaluated the effect of whole mix diet concentrate and roughage containing different proportions of millet straw and corn straw on the growth performance, blood metabolites, immune capacity, and carcass traits of lambs. The results proved the feeding effect of millet straw substituted for 50% of corn straw in fattening lambs.

**Abstract:**

As the byproduct of finger millet, millet straw is a new forage resource of ruminants. The effect of the combined utilization of millet straw with corn straw on fattening lamb production is seldom reported. The purpose of this study was to investigate the effect of different proportions of millet straw instead of corn straw on the growth performance, blood metabolites, immune response, meat yield, and quality of fattening lamb. Sixty-three-month-old healthy Small-Tailed Han sheep crossbred rams with an average initial weight of 19.28 ± 2.95 kg were randomly divided into four groups, with three replicates in each group and five lambs in each replicate. The replacement ratio of millet straw of each group (Group Ⅰ, Ⅱ, Ⅲ, Ⅳ) was 0%, 25%, 35%, 50% at the first stage (the first two months) and 0%, 20%, 28%, 40% in the second period (final two months), respectively. The experiment lasted 4 months 10 days of the pre-feeding period. The results indicated that the body weight gain and average daily gain of group Ⅱ were significantly higher than those of group Ⅰ and group Ⅳ (*p* < 0.05). The concentration of total protein in group Ⅳ was significantly increased compared to those of the other three groups at the second stage (*p* < 0.05), which proved that the protein synthesis metabolism capacity was improved with the addition of millet straw. The concentration of the plasma glutamic-oxalacetic transaminase and lactic dehydrogenase of lambs was significantly decreased in group Ⅱ (*p* < 0.05). The combination of millet straw and corn straw had no impact on the glucose, total cholesterol, and triglycerides metabolism (*p* > 0.05). There was no significant difference in the pre-slaughter weight, carcass weight, dressing percentage, loin-eye area, and GR value among the four groups (*p* > 0.05). Furthermore, the immune response and meat quality were not impacted by the different proportions of millet and corn forage diets. The results showed that the combined utilization of millet straw with corn straw could improve the blood biochemistry metabolism capability of fattening lambs. The replacement of 50% of corn straw with millet straw could improve the growth performance and be an application in fattening lamb production.

## 1. Introduction

Plenty of studies have confirmed that straw contains satisfactory protein and energy levels which could meet the requirements of animals and could be mixed with pasture for ruminant feeding [1,2,3,4,5,6,7]. Millet straw is a byproduct from the finger millet crop, which has been widely planted in arid and semi-arid areas of the world [8]. Therefore, millet straw is an important roughage forager source for ruminants in many areas.

Finger millet straw was already used in ruminant feeding, such as sheep [6,9,10,11] and heifers [12]. Crop straws often were specially treated by biophysical or chemical methods, such as sodium hydroxide treatment [13,14,15] the addition of urea to improve the digestibility and nutritional value [16,17,18]. Research reported that it is essential to provide an adequate amount of rumen-degradable nitrogen (RDN) for optimum microbial protein synthesis in the rumen on straw-based diets. The result indicated that the addition of RDN in diets based on finger millet straw could increase the optimum microbial protein synthesis (MPS), nitrogen capture efficiency (NCE), and nutrient utilization in Nellore rams [10]. Previously, studies have reported that the addition of degradable nitrogen could improve the microbial protein synthesis and digestibility of nutrients, enhancing the plane of nutrition in sheep fed on a finger millet straw-based diet [10,19,20]. Recent research has concluded that the urea molasses treatment could be taken as an option to improve the nutritive value of locally available lowland bamboo leaf hay and finger millet straw. Moreover, the urea molasses treatment improved the finger millet straw more than lowland bamboo leaf in most nutrients [6].

Furthermore, different kinds of hay match with the constituted diets often used in ruminants. The rapidly increasing human population and its demand for food production have caused the conversion of rangelands and pasturelands into cultivation land in some countries [21]. The change in land conversion is anticipated to proceed at an accelerating pace with population growth. A maximum utilization and systematic study on the associated effects of different crop straws was important for a large-population country. Hence, an amount of crop straws were utilized as ruminant fodder. The effects of replacing alfalfa hay (AH) with a mixture of cassava foliage silage and sweet potato vine silage (CSP) (1:1 on a dry matter (DM) basis) on the ruminal and intestinal nutrient digestion were investigated in sheep. The results demonstrated that replacing AH with the CSP reduced the ruminal N degradation, as well as the digestion of ruminal neutral detergent fiber and intestinal N [3]. The effects of different forage grasses on animals were ultimately reflected in their productive performance. However, the effects of the mixture diet on the production performance and meat quality were not investigated in this experiment.

In rural areas, corn straw is abundant for sheep as a fodder in China. However, little is known about the effect of millet straw associated with corn straw on fattening lambs. The objective of this study was to evaluate the effect of whole mix diet concentrate and roughage containing different proportions of millet straw and corn straw on the growth performance, blood metabolites, immune capacity, and carcass traits of lambs.

## 2. Material and Methods

The experiment was conducted at the farm of Guo liang, at the Xuanhua county of Hebei province in China. The experiment site is characterized by an average annual temperature of 8 °C, and an average elevation of approximately 1200 m. The studies were conducted at Hebei Agricultural University. All the animal manipulations were approved by the Ethics Committee of Experimental Animal of Hebei Agricultural University (2020087).

### 2.1. Animals, Diets, and Experiment Design

Sixty-three-months-old healthy Small-Tailed crossbred rams with an average initial weight of 19.28 ± 2.95 kg were completely randomly divided into four groups, with three replicates in each group and five lambs in each replicate. The diets, formulated based on the NRC (2007) for an average daily gain of 200 g and supplied as a complete mix, were composed of ingredients including corn, soybean meal, mineral supplement, and roughage containing millet straw, of which the proportion was 0%, 25%, 35%, 50% at the earlier stage (the first two months) and 0%, 20%, 28%, 40% in the later period (final two months). The roughage-to-concentrate ratio was 50:50 at the first stage and 40:60 in the second period (Table 1).

The experimental assay lasted 4 months and was preceded by a 10-day adaptation period, during which vaccination, deworming, management, and diet adaption procedures were performed. The average daily gain (ADG) was calculated as the initiation and terminal weight during the experiment. The lambs were fed twice daily, at 8:00 and 16:00 h.

### 2.2. Blood Profiles

In the mornings (before feed distribution) of the first days of the finished adaption and the day of the terminal experiment, blood samples were taken from the jugular vein of five individuals of each group in tubes. These samples were immediately centrifuged (3000 rpm at 4 °C) and the plasma was recovered and stored at −20 °C until it was analyzed. Immunoglobulin G (Ig G), immunoglobulin M (Ig M), interferon-γ (IFN-γ), interleukin-2 (IL-2), interleukin-6 (IL-6), glutathione peroxidase (GSH-Px), total antioxidant capacity (T-AOC), and complement C3 and C4 were measured by an enzyme-linked immunosorbent assay. The plasma metabolites, including total protein (TP), glutamic-pyruvic transaminase (GPT), glutamic-oxalacetic transaminase (GOT), glucose, urea, total cholestenone, triglyceride, and lactic dehydrogenase, were measured using an automated analyzer for biochemistry (Lexington, MA, USA) by dedicated Kits (Sigma-Aldrich, Shanghai, China).

### 2.3. Carcass Traits

At the termination of the trial, the animals were slaughtered according to the current norms of normative instruction of the ministry of agriculture and country of China, and their carcasses were trimmed and weighed to determine the dressing percentage. Five individuals of each group of lambs were slaughtered in a commercial abattoir, and the body weight at slaughter (BWS) was recorded. The longissimus dorsi was collected from each carcass for a meat quality analysis. The GR values named steak piece in mutton standard of China (NY/T630-2002), which indicate the fat content of carcass, were evaluated by making a cross section between the 13th thoracic vertebra and the first lumbar vertebra, allowing the exposure of the cross-section of the loin from the right half-carcass. The loin-eye area was measured with a transparent, gridded standard template, in which each square corresponded to 1 cm^2^ [22].

### 2.4. Meat Quality

The pH1 and pH24 of the muscle were respectively measured by a PH instrument at 1 h and 24 h keep in cold storage after slaughter. The longissimus lumborum were vacuum packaged and frozen at −20 °C in a domestic freezer until the analysis. After removing the subcutaneous fat, the samples were cooked in a temperature-controlled water bath to an internal temperature of 75 °C. The cooked meat percentage was calculated from the difference in the weight of the raw and cooked samples and was expressed as a percentage of the initial weight.

### 2.5. Statistic Analysis

A one-way ANOVA in SPSS20.0 program (IBM Corp., Armonk, NY, USA) was used to determine the significance of the effect of four diets on the lambs’ growth performance, plasma metabolites, immune response, and carcass quality characteristics. Duncan’s multiple range test was also used to evaluate the significance of the difference. All the effects were tested for statistical significance (*p* < 0.05), and significant effects were reported in tables. When significant differences were found (*p* < 0.05), the *t*-test was used to locate significant differences between the means.

## 3. Results

### 3.1. Growth Performance

The total gain weight (TGW) and average daily gain (ADG) were affected by the dietary treatment (Table 2). The ADGs of group Ⅱ were higher than those of group Ⅰ and Ⅳ (*p* < 0.05), which improved 11.27% compared to group Ⅲ.

### 3.2. Blood-Biochemical Parameters

The concentrations of blood metabolites were affected by the diets. There were apparent differences in the concentrations of TP, GPT, GOT, glucose, urea, and lactic dehydrogenase at the final phase between the different diet groups. At the terminal stage, the concentration of TP in group Ⅳ was significantly increased compared to those of group Ⅲ (*p* < 0.05), group Ⅰ, and group Ⅱ, respectively (*p* < 0.01). The plasma GPT and GOT concentrations of group Ⅱ were significantly lower than those of group Ⅳ (*p* < 0.05). There was no significant difference in the serum glucose, total cholestenone, and triglyceride in each group (*p* > 0.05). The serum urea concentrations of group Ⅱ were obviously lower than those of group Ⅳ (*p* < 0.01) and group Ⅲ (*p* < 0.05). The serum lactic dehydrogenase contents of group Ⅱ were significantly lower than those of the other three groups (*p* < 0.05). The provision of diet composition had no impact on the total cholesterol and triglycerides (*p* > 0.05) (Table 3). Any apparent difference was not found in the immune capacity between diet groups (*p* > 0.05) (Table 4).

### 3.3. Immunity, Carcass Trait, and Meat Quality

All the indexes related to immune capacity were not statistically different among the treatments (*p* > 0.05). There were no effects of diets on the BWS, carcass weight, dressing percentage, GR value, and loin-eye area (*p* > 0.05) (Table 5). The meat quality was not impacted by the diets (*p* > 0.05) (Table 6).

## 4. Discussion

This study is the first to investigate the effect of finger millet straw associated with corn straw in the diet on the growth performance and meat quality as well as the plasma metabolites and immune capacity in fattening lambs. The aim of this study was to provide evidence that it is more practicable to use millet straw as a basic forage, similar to alfalfa in animal production, in regions where the finger millet crop is widely planted. The results indicated that diets with a mixture of millet straw and corn straw did not affect the final body weight. However, the differences in the daily weight gain and total weight gain of the lambs can be explained by the digestive absorption, which may be different among the diets and which is the nutritional factor of greatest influence on weight gain [23]. Similar results are reported for finger millet straw as a basic diet with different protein sources, which can improve the feed utilization, body weight gain, and digestibility in lambs [24]. Furthermore, dry matter intake is another factor influencing weight gain. It was observed that as the dry matter intake decreases, animals fed peanut cake as a replacement for soybean meal also showed lower weight gains [25]. The previous study indicated that certain amounts of digestible organic matter intake or that apparently digested in the rumen can improve the plane of nutrition in sheep fed on a finger millet straw-based diet [19].

The plasma concentrations of total protein, urea, and hepatic enzymes were correlated with the metabolism and animal health. At the terminal stage, the concentration of total protein in group Ⅳ was significantly increased than those of the other three groups (*p* < 0.05), which demonstrated that the protein synthesis metabolism capacity improved with the millet straw addition proportion. The plasma GOT concentration of animals were significantly decreased in group Ⅱ (*p* < 0.05), which illustrated that the impact of the group Ⅱ diet on the liver was lower than that of the other groups’ diets. However, our results were different from the latest research, in that the plasma concentrations of total protein, urea, and hepatic enzymes were not affected by the diet containing corn distiller dried grains with solubles (DDGS; 18%), dried citrus pulp (DCP; 18%), and exhausted olive cake (EOC; 8%) in light lambs [26]. The result of the lower plasma lactic dehydrogenase level indicated that the anti-stress capability of lambs was enhanced in group Ⅱ. The provision of diet composition had no impact on the glucose, total cholesterol, and triglyceride metabolism in this study. It has been reported that the concentrations of triglycerides can be influenced by essential oil supplementation via changes in feed intake [27], and that the lack of changes in triglyceride or cholesterol content may contribute to the failure of essential oil to cause alterations in DMI [28].

The capacity of the immune system is vital for health during fattening, of which the level not only affects the daily weight gain but also the disease incidence. In the last few years, several studies have investigated the strong relation between nutrition and immune response in livestock production, particularly in dairy cows [29,30,31] and sheep [32,33]. We measured an index of the immune system for appreciating the mixture diets with the same nutrition concentration. The results of all the index were not statistically different among the treatments, and illustrate that different compositions of straw did not affect the immune function. Meanwhile, the serum immunoglobulin levels (IgG, IgM, and IgE) and proinflammatory cytokine levels (TNF-α, IL-1, and IL-6) significantly increased on days 60 and 120 (*p* < 0.05) in a diet based on palm oil [33]. The previous study demonstrated that the diet supplementation linseed and quinoa enhanced the cell-mediated immune response of lambs [32]. Jiang et al. reported that a dietary curcumin supplement can promote lipid metabolism, antioxidant capacity, and immune response [34].

The body weight at slaughter, carcass weight, and dressing percentage, which were important evaluation indicators of carcasses, have different economic values. The first two indexes were the live body weight before slaughter and the size of the carcass without the head, fur, viscera, and tail. The dressing percentage, percentage of carcass weight, and BWS was an essential parameter for the evaluation of the carcass efficiency of live sheep. The dressing percentage, GR value, and loin-eye area were used for the determination of the meat yield of carcasses. The results regarding the above three indexes were not influenced by the diets in this experiment. The carcass traits were not significantly different among the four treatments (Table 5). This is in agreement with the present study, which did not observe any difference in the carcass morphometric measurement among the different diets [35] and oil seed meal diet [23].

Some studies about diet nutrition with meat quality were conducted in sheep [36,37,38,39,40,41,42,43,44,45]. Protes et al. (2018) reported the decreased shear force of samples from lambs fed with soybean silage compared to sorghum. The finding indicated that the PH of meat could be a good indicator of pork quality and related to factors influencing the pork eating quality [46]. It was reported that the release of adrenaline led to an ante mortem glycogen breakdown and, as a consequence, to a lower pH value [47]. In addition, the post-mortem pH (24 h) was positively correlated with the water holding capacity, but negatively correlated with the meat color, protein content, drip loss, and cooking loss [48]. Our findings manifested that all the meat quality parameters measured were no statistically different among the treatments.

No effect of the levels of millet straw in the substitution of corn straw meal was verified in any of the carcass traits and meat quality as well as immune capacity. These results can be explained by the fact that the animals received the experimental diets, which have no negative influence on the above aspects during the fattening period.

## 5. Conclusions

The addition of millet straw in a whole mixed diet could improve the blood biochemistry metabolism capability. In addition, the average daily gain was maximum in the lambs fed a diet with forage composed of 50% millet straw and 50% corn stalk, which presented the idea that millet straw could be applied as a new forage resource in fattening lambs.

## Figures and Tables

**Table 1 animals-10-01285-t001:** Proportion and chemical composition of the experimental diets.

Item	Firs Stage				Second Period			
	Group Ⅰ	Group Ⅱ	Group Ⅲ	Group Ⅳ	Group Ⅰ	Group Ⅱ	Group Ⅲ	Group Ⅳ
Ingredient								
Corn	25.00	25.00	25.00	25.00	39.00	39.00	39.00	39.00
Til meal	10.75	10.75	10.75	10.75	9.00	9.00	9.00	9.00
Wheat bran	5.00	5.00	5.00	5.00	3.60	3.60	3.60	3.60
Soybean meal	7.00	7.00	7.00	7.00	6.00	6.00	6.00	6.00
Til meal	0.50	0.50	0.50	0.50	0.30	0.30	0.30	0.30
Salt	0.50	0.50	0.50	0.50	0.60	0.60	0.60	0.60
Calcium bicarbonate	0.65	0.65	0.65	0.65	0.78	0.78	0.78	0.78
Calcium hydrophosphate	0.10	0.10	0.10	0.10	0.12	0.12	0.12	0.12
Premix	0.50	0.50	0.50	0.50	0.60	0.60	0.60	0.60
Corn straw	50.00	25.00	15.00	0.00	40.00	20.00	12.00	0.00
Millet straw	0.00	25.00	35.00	50.00	0.00	20.00	28.00	40.00
Total	100.00	100.00	100.00	100.00	100.00	100.00	100.00	100.00
Nutrient levels								
Digestible energy (MJ/Kg)	10.55	10.53	10.53	10.52	11.40	11.39	11.38	11.38
Crude Protein (%)	12.99	13.06	13.32	13.37	11.88	12.02	12.50	12.44
Neutral detergent fiber (%)	57.38	55.14	56.60	54.44	45.45	45.97	46.02	44.49
Acid detergent fiber (%)	33.88	33.44	32.01	32.99	25.41	27.46	28.31	26.95
Calcium (%)	0.69	0.67	0.68	0.64	0.57	0.53	0.51	0.53
Total phosphorus (%)	0.24	0.23	0.23	0.22	0.20	0.20	0.19	0.20

**Table 2 animals-10-01285-t002:** The effect of different diets on the lambs’ growth performance.

Item	Group Ⅰ	Group Ⅱ	Group Ⅲ	Group Ⅳ
FBW (kg)	42.04 ± 2.88	46.15 ± 3.88	44.49 ± 3.81	42.26 ± 5.22
TGW (kg)	23.07 ± 2.67 ^b^	27.36 ± 4.02 ^a^	24.51 ± 3.52 ^ab^	23.22 ± 4.14 ^b^
ADG (g)	192.21 ± 22.27 ^b^	227.99 ± 33.49 ^a^	204.22 ± 29.31 ^ab^	193.46 ± 34.48 ^b^

FBW: final body weight is the terminal weight during the fattening period; ADG, average daily gain of lambs during fattening experiment; TGW: total gain weight is the average gain in weight of all the lambs from initiation to termination during the experiment. Superscript letters shared in common between the groups indicate no significant difference (*p* > 0.05), and superscript letters shared in different between the groups indicate significant difference (*p* < 0.05).

**Table 3 animals-10-01285-t003:** Effect of the different diets on the plasma metabolites in lambs.

Item	Stage	Group Ⅰ	Group Ⅱ	Group Ⅲ	Group Ⅳ
TP, g/L	Initiation	54.10 ± 3.25	55.02 ± 2.54	55.08 ± 1.29	53.03 ± 1.06
	Final	30.75 ± 4.47 ^Bb^	31.22 ± 4.32 ^Bb^	33.27 ± 8.35 ^ABb^	43.25 ± 2.94 ^Aa^
GPT, U/L	Initiation	15.00 ± 5.5	15.75 ± 1.71	13.25 ± 3.77	12.50 ± 3.00
	Final	8.00 ± 2.12 ^ab^	7.75 ± 1.89 ^ab^	6.50 ± 1.91 ^b^	10.80 ± 2.49 ^a^
GOT,(U/L)	Initiation	126.00 ± 22.79	117.00 ± 9.13	127.25 ± 27.55	123.00 ± 24.55
	Final	79.2 ± 16.16 ^ab^	66.20 ± 16.95 ^a^	67.00 ± 25.16 ^ab^	95.67 ± 24.74 ^b^
Glucose, mmol/L	Initiation	4.40 ± 0.46 ^Aa^	4.07 ± 0.73 ^Aba^	3.76 ± 0.26 ^Abab^	3.36 ± 0.41 ^Bb^
	Final	2.94 ± 0.32	2.63 ± 0.49	3.29 ± 1.15	3.78 ± 0.79
Urea, mmol/L	Initiation	7.61 ± 2.35 ^Aa^	10.04 ± 1.19 ^Aba^	13.46 ± 2.52 ^Bb^	13.31 ± 1.51 ^Bb^
	Final	4.74 ± 1.04 ^Abab^	3.68 ± 0.60 ^Bb^	5.15 ± 0.56 ^Aba^	5.81 ± 1.34 ^Aa^
Total cholesterol, mmol/L	Initiation	1.12 ± 0.19	1.22 ± 0.08	1.15 ± 0.21	1.22 ± 0.21
	Final	0.73 ± 0.26	0.75 ± 0.22	0.92 ± 0.25	0.93 ± 0.26
Triglycerides, mmol/L	Initiation	0.37 ± 0.12	0.36 ± 0.07	0.27 ± 0.04	0.32 ± 0.05
	Final	0.21 ± 0.04	0.21 ± 0.04	0.27 ± 0.05	0.28 ± 0.06
Lactic dehydrogenase, U/L	Initiation	454.67 ± 32.88	429.60 ± 23.73	455.00 ± 54.83	420.75 ± 50.08
	Final	407.75 ± 57.30 ^Aab^	283.5 ± 34.38 ^Bc^	342.33 ± 62.95 ^Abb^	467.67 ± 34.56 ^Aa^

TP, total protein; GPT in plasma, glutamic-pyruvic transaminase; GOT, glutamic-oxalacetic transaminase; GPT and GOT were important liver transaminases for metabolism. Superscript letters shared in common between the groups indicate no significant difference (*p* > 0.05), superscript lowercase (*p* < 0.05) and capital letters (*p* < 0.01) shared in different between the groups indicate significant difference.

**Table 4 animals-10-01285-t004:** Effect of different diets on the immune capacity of the lambs.

Item	Group Ⅰ	Group Ⅱ	Group Ⅲ	Group Ⅳ
Ig G, ug/mL	537.65 ± 60.62	564.70 ± 86.30	660.42 ± 93.39	567.84 ± 90.41
Ig M, ug/mL	371.21 ± 65.45	378.20 ± 45.55	385.31 ± 40.56	336.45 ± 20.00
IFN-γ, pg/mL	587.05 ± 121.83	535.54 ± 69.87	556.35 ± 92.87	609.39 ± 126.91
IL-2, pg/mL	710.23 ± 85.85	670.93 ± 192.32	701.85 ± 161.57	656.76 ± 63.71
IL-6, pg/mL	87.22 ± 14.89	99.93 ± 12.72	95.81 ± 21.30	80.42 ± 18.8
GSH-Px, pg/mL	1240.22 ± 362.02	1294.02 ± 358.65	1252.64 ± 274.79	1343.53 ± 389.56
T-AOC, U/mL	19.55 ± 4.29	20.40 ± 3.08	19.64 ± 5.26	21.64 ± 4.80
C3, ug/mL	35.86 ± 10.32	36.61 ± 16.82	41.91 ± 5.39	40.13 ± 8.75
C4, ug/mL	12.81 ± 1.31	13.47 ± 2.09	12.50 ± 2.35	12.75 ± 3.53

Ig G, immunoglobulin G; Ig M, immunoglobulin M; Ig G and Ig M were members of the immunoglobulin family; IFN-γ, interferon-γ. IL-2, interleukin-2; IL-6, interleukin-6; IL-2 and IL-6 were cytokines correlated with immunity; GSH-Px, glutathione peroxidase; T-AOC, total antioxidant capacity; GSH-Px and T-AOC represent the oxidation resistance of the organism; C3, complement C3; C4, complement C4. C3 and C4 were important members of the complement system in immune capacity.

**Table 5 animals-10-01285-t005:** Effects of different diets on the carcass traits.

Item	Group Ⅰ	Group Ⅱ	Group Ⅲ	Group Ⅳ
BWS, kg	42.58 ± 3.94	41.47 ± 5.74	43.52 ± 4.82	44.90 ± 5.61
Carcass weight, kg	20.56 ± 3.25	20.78 ± 3.18	20.93 ± 2.20	21.69 ± 3.26
Dressing percentage, %	48.58 ± 3.65	49.51 ± 2.01	48.12 ± 1.01	48.19 ± 1.98
GR value, mm	11.20 ± 0.52	11.11 ± 0.41	11.67 ± 0.77	11.89 ± 0.63
Loin-eye area, cm^2^	15.71 ± 1.51	16.81 ± 3.83	16.92 ± 2.31	16.68 ± 3.03

BWS, body weight at slaughter.

**Table 6 animals-10-01285-t006:** Effects of different diets on the meat quality.

Item	Group Ⅰ	Group Ⅱ	Group Ⅲ	Group Ⅳ
pH1	6.29 ± 0.28	6.09 ± 0.24	6.14 ± 0.16	6.14 ± 0.12
pH24	5.60 ± 0.25	5.79 ± 0.16	5.71 ± 0.14	5.72 ± 0.27
Cooked meat percentage, %	57.73 ± 2.80	57.64 ± 3.65	58.54 ± 3.64	56.80 ± 2.23
Water lose rate, %	50.79 ± 2.39	50.33 ± 2.01	50.21 ± 2.45	50.15 ± 2.28
Shear force, N	54.28 ± 3.61	50.15 ± 9.76	53.60 ± 3.23	49.98 ± 4.50
Marbling score	1.80 ± 0.45	2.00 ± 0.70	1.80 ± 0.84	2.60 ± 0.89

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
