# Peer review of "Effects of Diets Containing Finger Millet Straw and Corn Straw on Growth Performance, Plasma Metabolites, Immune Capacity, and Carcass Traits in Fattening Lambs"

_animals, 2020, doi:10.3390/ani10081285_

Round 1

Reviewer 1 Report

General comment

The research is interesting. However, the introduction and the discussion could be improved. The language is good and only slight corrections would be required.

Specific concerns:

Line 13: Replace "was" with "been".

Line 20: Replace "In this study, the purpose of this experiment" with "The purpose of this study".

Line 23: Replace "60" with "sixty".

Line 25: It is not clear how many lambs were involved in this study for each group. Clarify better the experimental design. Indicate the groups (Group I, II, III, IV) near the percentage of millet straw integration.

Line 27: Add the verb.

Line 46-65: The introduction is not sufficient and it could be improved.

Line 71-72: "….were completely randomly divide into four groups with three replicates in each group and five lambs in each replicate….". The experimental design is not clear; describe better. What is meant for replicate?

Line 73-78: The percentage of millet straw integration were not the same reported in the abstract. Describe better the diet. How long did the earlier stage last? and the later period?

Line 84: Add "was".

Line 123: Replace Total gain weight "(TGA)" with "TGW".

Insert in tables footnotes a more precise description of superscript letters.  

Make the font of significance uniform in the whole manuscript (e.g. P < 0.05). 

Table 2, 3, 4, 5: Delete the superscript "a" letter near Item.

Line 181-183: Delate the sentence "The aim of this study was to evaluate the effects of supplementation based on millet straw, corn straw and their combination on fatten lambs."

Line 167-177: Improve the discussion of these data.

Author Response

We thank you for the positive evaluation of our manuscript (Manuscript ID: 862362) for publication in Animals. We considered your comments carefully and revised the manuscript accordingly. The point-by-point responses to each comment are explained in detail below.

Reviewer 2 Report

Manuscript ID: animals-862362

Recommendations of Reviewer

Title:

I suggest to change the title to: “Effect of diets contained finger millet straw and corn straw on growth performance, plasma metabolites, immune capacity and carcass traits in fatten lambs”.

In summary: Please revise english language.

In material and methods

Include time of measurement pH1 and pH2 of muscle.

In Discussion

Add the importance of measuring pH1 and pH2.

Please add more references about sheep diets similar to the used in this study.

In general the, english language of the whole mansucript should be careful revised.

Author Response

(The authors gave the same response as above.)

Round 2

Reviewer 1 Report

The manuscript has been revised according to the suggestions and comments. 

Reviewer 2 Report

From my view point, the manuscript is correct.

All suggested correction were done.

Thank you.